# Simultaneous EEG-fNIRS study of visual cognitive processing: ERP analysis and decision-related hemodynamic responses in healthy adults

Tolga Turay[1], Onur Erdem Korkmaz[1,2], Ebru Ergün[3]*

**1** Department of Electrical and Electronics Engineering, Atatürk University, Erzurum, Turkey, **2** Sports Sciences Application and Research Center, Atatürk University, Erzurum, Turkey, **3** Department of Electrical and Electronics Engineering, Faculty of Engineering and Architecture, Recep Tayyip Erdogan University, Rize, Turkey

* ebru.yavuz@erdogan.edu.tr

## Abstract

This study explores the neural and hemodynamic underpinnings of intentional memory processing through a multimodal approach combining electroencephalography (EEG) and functional near-infrared spectroscopy (fNIRS). Data were drawn from a publicly available dataset in which participants viewed visual scenes and decided whether to remember them, enabling classification into four experimental conditions based on motivation and subsequent memory performance: Want to Remember and Remembered (RR), Want to Remember but Forgot (RF), Did Not Want to Remember but Remembered (FR), and Did Not Want to Remember and Forgot (FF). EEG analyses focused on event-related potentials (ERPs) during the first second following stimulus presentation. The RR and RF conditions showed enhanced ERP amplitudes, particularly in parietal and occipital channels, peaking around 300 ms post-stimulus. Time-frequency analysis using wavelet transform further revealed greater theta and low alpha power in the RR and RF conditions, again especially in parietal and occipital regions. fNIRS analysis examined hemodynamic responses during the subsequent 9-second decision period. While visual inspection revealed variability in oxygenated hemoglobin (HbO) levels across channels and conditions, statistical analyses using Cohen's D and one-way ANOVA did not identify any significant differences between the conditions ($p > 0.05$). These findings suggest that while EEG metrics capture early, intention-driven neural dynamics, fNIRS may reflect more distributed and variable patterns of cognitive engagement. The integration of EEG and fNIRS provides a comprehensive framework for investigating cognitive motivation and memory, highlighting the temporal and spatial signatures of intentional memory processing.

**Data availability statement:** The dataset used in this study is publicly available at the following link: https://data.mendeley.com/datasets/z92nw4n73t/3.

**Funding:** This research was financially supported by the Recep Tayyip Erdogan University Development Foundation (Grant number: 02025004007391). We sincerely appreciate their support, which contributed significantly to the completion of this study. The authors gratefully acknowledge the support of the Scientific Research Projects Coordination Unit of Atatürk University under Project Number FBA-2025-15082.

**Competing interests:** The authors declare no conflicts of interest.

## Introduction

Motivation is the set of psychological processes and states that prompt individuals to take action toward a goal for internal or external reasons. It can be defined as a person's desire, need, or wish to perform an action. Motivation directs an individual's energy, helps sustain their efforts, and plays a significant role in achieving their goals [1–4]. In the field of educational psychology, motivation is critically important in enhancing students' engagement in learning processes and increasing their efficiency. By boosting students' participation in classes, motivation helps them understand and remember information better. It also ensures that students exert the necessary effort to achieve specific academic goals. When faced with challenges and failures, motivation enables students to persist in their efforts without giving up, helping them overcome obstacles. Motivated students can better regulate their own learning processes, contributing to the development of self-regulation skills such as self-motivation and self-monitoring [5–9]. Lastly, motivation increases students' study time and productivity, generally resulting in higher academic achievement. Therefore, it is essential for teachers and educators to continually develop their methods and strategies for motivating students, as this helps make educational environments more effective and efficient. Thus, measuring and assessing motivation, particularly from a cognitive perspective, is highly important. Proper understanding and monitoring of motivation play a critical role in enhancing the effectiveness of educational strategies and maximizing student success [7,10–12] .

Cognitive motivation has been a significant area of research, particularly in understanding how various motivational factors influence cognitive processes and learning outcomes. A variety of studies have explored the interplay between cognitive motivation and factors such as intrinsic motivation, emotional motivation, and cognitive load, providing insights into how these elements can enhance or hinder learning. Intrinsic motivation has been shown to play a crucial role in cognitive processing and memory retention. Yang's research highlights that individuals who possess a genuine curiosity and intrinsic desire for knowledge exhibit enhanced cognitive processing, leading to improved memory retention and recall [13]. This aligns with findings from Nakagami et al., who demonstrated that incorporating intrinsically motivating instructional methods into challenging cognitive tasks can significantly enhance learning outcomes and self-efficacy [14]. Furthermore, Ghanizadeh and Jahedizadeh emphasize the importance of self-regulated learning, where learners actively engage in monitoring and regulating their cognition and motivation, thereby positively influencing their academic achievement [15]. Emotional motivation also significantly impacts cognitive flexibility and strategy use. Wang et al. found that high emotional motivation can restrict cognitive processing, which may impede cognitive flexibility, particularly under varying task loads [16]. This suggests that while motivation is essential, its nature (approach vs. avoidance) can lead to different cognitive outcomes. In the context of adult education, Francois's study indicates that cognitive interest serves as a strong motivator for non-traditional students, further emphasizing the role of cognitive factors in educational settings [17]. Moreover, the relationship between cognitive motivation and

academic performance has been explored in various contexts. Sereda et al. identified educational and cognitive motives as the strongest predictors of students' Grade Point Averages, reinforcing the idea that a need for cognition is closely linked to learning outcomes [18]. Similarly, the research by Vallet et al. suggests that motivation mediates the relationship between cognitive reserve and cognitive performance, indicating that higher motivation can help mitigate cognitive decline [19]. In addition to intrinsic and emotional motivations, cognitive load has emerged as a critical factor influencing learning motivation. Studies by Deleeuw and Mayer illustrate that managing cognitive load effectively can enhance motivation and learning outcomes, as students are more likely to engage deeply with the material when cognitive resources are optimally allocated [20]. This is echoed in the work of Rossi et al., which highlights the importance of emotional, behavioral, and cognitive dimensions in student engagement, particularly in online learning environments [21]. In conclusion, the body of research on cognitive motivation underscores its multifaceted nature, revealing that intrinsic motivation, emotional factors, and cognitive load all play integral roles in shaping cognitive processes and learning outcomes. These insights are crucial for educators and policymakers aiming to foster effective learning environments that enhance student motivation and achievement.

Several electroencephalography (EEG) studies have examined how motivation can impact various brain regions, depending on the underlying motives. For instance, van der Ven et al. focused on the N400 event-related potential (ERP) component during a reading task where rewards were contingent upon performance. Their findings indicated that electrodes located in the central and parietal regions, specifically C3, Cz, C4, CP1, CP2, P3, Pz, and P4, were significantly affected by motivational factors [22]. Similarly, Ma et al. investigated the influence of motivation arising from challenging tasks by analyzing the mean amplitude of stimulus-preceding negativity, another ERP component, in 2017. Their results revealed that the difficulty level of the task impacted frontal area electrodes, including F4, F6, F8, FC4, FC6, and FT8 [23]. In a different approach, Jin et al. explored the effects of interesting versus boring tasks through the P300 and feedback-related negativity (FRN) ERP components. They identified significant differences between the two conditions in electrodes F1, Fz, F2, FC1, FCz, FC2, C1, Cz, and C2 for the FRN, and in C1, Cz, C2, CP1, CPz, CP2, P1, Pz, and P2 for the P300 component [24]. Lastly, Brydevall et al. conducted a study in 2018 focusing on an information-seeking task driven by curiosity, utilizing the FRN ERP component. Their research highlighted the influence of motivation on electrodes Fpz, AFz, Fz, FCz, and Cz [25]. In a study conducted in the literature, researchers explored the intricate relationship between motivation and task performance, emphasizing the varying effects of different motivational factors on distinct brain regions. Traditional methodologies for analyzing these effects often necessitate a substantial number of EEG electrodes, which can lead to increased costs and user inconvenience. To mitigate this challenge, the study introduced an innovative approach utilizing Temporal Association Rule Mining (TARM) to examine the connections between attention and memory-related brain areas influenced by cognitive motivation. Furthermore, the researchers applied the Artificial Bee Colony algorithm in conjunction with the Central Limit Theorem to optimize the parameters of TARM. The results indicated that the motivation effect could be effectively identified using only the FCz and P3 electrodes, achieving an average classification accuracy of 74.5% across individual tests. This novel method not only reduces the number of required electrodes but also maintains adequate accuracy, suggesting its practical applicability for performance evaluation in real-world contexts [26].

The integration of EEG and functional near-infrared spectroscopy (fNIRS) presents a promising avenue for advancing our understanding of cognitive motivation and its effects on brain activity. EEG provides high temporal resolution, allowing for the precise tracking of neural dynamics, while fNIRS offers superior spatial resolution and robustness to noise, making it an ideal complement to EEG. Although EEG studies have been extensively conducted in the realm of cognitive motivation, to the best of our knowledge, fNIRS signals have not yet been utilized in this specific context. By employing hybrid EEG-fNIRS signals, this research aims to explore cognitive motivation from a novel perspective, leveraging existing datasets to provide a comprehensive analysis of brain activity related to motivation. This innovative approach not only enhances the understanding of the neural correlates of motivation but also contributes to the broader field of cognitive neuroscience by offering new insights into the interplay between electrophysiological and hemodynamic responses during

cognitive tasks. The findings from this study may pave the way for future research that further investigates the complexities of motivation and cognition through multimodal neuroimaging techniques.

## Materials and methods

### Data Set

This study utilizes the "A simultaneous EEG-fNIRS dataset of the visual cognitive motivation study in healthy adults" dataset created by Phukhachee and colleagues. This open-access dataset encompasses data collected from participants performing a visual cognitive motivation task, where EEG and fNIRS measurements were taken simultaneously [27]. The dataset was primarily aimed at gaining a deeper understanding of motivation through the intensive monitoring of physiological responses. For inclusion in the dataset, adult participants without any prior history of visual perception or memory disorders were selected. The sample size for this study was determined using the Lemeshow method [28]. Additionally, parameters derived from a similar cognitive motivation task studied by Yoo and colleagues using fMRI measurements were incorporated [29].

Data was collected from 16 participants who signed informed consent forms. The participants were composed of 14 males and 2 females, aged between 21–37 years. All experimental procedures were approved by the Ethics Committee of the Faculty of Information Science and Electrical Engineering at Kyushu University.

The dataset includes EEG and fNIRS signals measured simultaneously during a cognitive motivation task. The experiment was divided into two parts: a cognitive motivation task and a recognition test. Brain activity signals were collected only while participants were engaged in the cognitive motivation tasks. Participants could freely choose whether they wanted to remember the scenic stimulus. The results from the recognition test were used to determine if participants could later recognize the scene. The experiment's trials can be categorized into four groups based on motivation and subsequent recognition: Want to Remember and Remembered (RR), Want to Remember but Forgot (RF), Did Not Want to Remember but Remembered (FR), and Did Not Want to Remember and Forgot (FF).

The visual cognitive motivation experiment consisted of two parts: a cognitive task and a recognition test. In the cognitive task, participants were shown random and unique visual stimuli of indoor or outdoor scenes taken from the Scene UNderstanding database. Each stimulus was displayed on the screen for 3 seconds, referred to as the attention span. Afterward, the screen was replaced with a fixation cross for 9 seconds, allowing participants to freely decide whether to remember the scene. This period is referred to as the decision time. During this time, participants responded by clicking the left mouse button if they wanted to remember the scene, and the right button if they did not. The 9-second decision time, while longer than typically required for EEG experiments, was adjusted to accommodate the measurement of hemodynamic responses with fNIRS. This extended decision time might reveal interesting features in fNIRS modulation. After the decision time, the screen was switched to another stimulus scene. A random short delay was added between the decision time and the appearance of the next stimulus to mitigate the repetition suppression effect. The entire procedure is depicted in Fig 1.

The cognitive task concluded with 250 scenes. With this setup, the cognitive experiment took approximately 50 minutes per participant, excluding setup time and short breaks requested by participants. After a 10-minute rest, the scene recognition test commenced. In the recognition test, a total of 500 scenes were presented one by one, including 250 from the cognitive experiment and 250 new scenes. Participants indicated whether they recognized the scenes from the cognitive task. This recognition test was used to verify the influence of motivation during the experiment. There was no time limit during this test, and brain signals were not measured.

During the cognitive experiment, EEG data were collected using a Nihon Kohden Neurofax EEG-1100 system with a sampling frequency of 500 Hz. fNIRS signals were obtained with a Hitachi ETG-7100 fNIRS device at a sampling frequency of 10 Hz. The recording locations for both modalities are shown in Fig 2. This setup allowed for the simultaneous

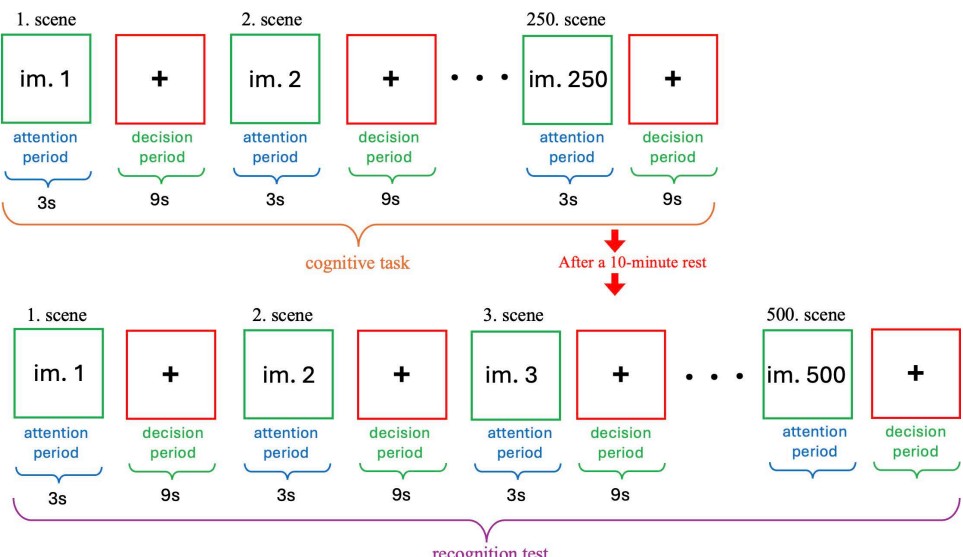

**Fig 1. Motivation-based cognitive experimental procedure for simultaneous EEG-fNIRS data collection.**

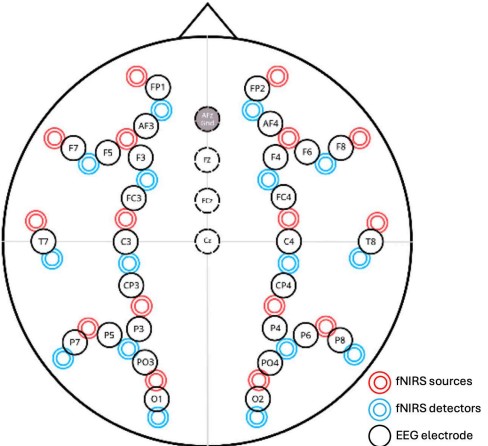

**Fig 2. Electrode and optode placement layout for EEG and fNIRS recording.**

measurement of electrophysiological and hemodynamic data at the same locations. The arrangement features black circles representing EEG electrodes, while red and blue circles indicate fNIRS sources and detectors, respectively.

## Data processing

In this study, the data processing procedure was conducted in two distinct phases: one for EEG signals and the other for fNIRS signals. Due to the inherent temporal resolution of EEG signals, they are well-suited for ERP analysis. Consequently, the attention period of the study focused on analyzing the ERP waves that occurred in the EEG signals following the presentation of an image. On the other hand, the spatial resolution of fNIRS signals is notably robust, which facilitated the analysis of fNIRS signals during the decision period. This structured approach allowed for a comprehensive examination of the neural mechanisms underlying each phase of the cognitive task.

**EEG data processing**

In the EEG data processing stage of our study, the segments were generated based on the responses provided by the participants as follows: 526 segments for the FF block, 1092 segments for the FR block, 332 segments for the RF block, and 1092 segments for the RR block. The numbers of FF, FR, RF, and RR segments created based on participant responses are shown in Table 1. Considering that the number of segments averaged in ERP analysis can influence the amplitude of the ERP, the lowest segment count from the RF block, 332, was employed to mitigate bias. This number was also randomly selected from the larger FF, FR, and RR blocks to compute the channel-based grand average ERP signals. The dataset providers supplied a 500 ms pre-stimulus EEG epoch and a 1000 ms post-stimulus EEG epoch. The pre-stimulus EEG segment was utilized as the baseline, from which noise was subtracted from the 1000 ms post-stimulus EEG segment to perform baseline noise correction.

In the EEG data processing stage, wavelet analysis was also conducted. During the wavelet transform step, average ERP signals obtained from all channels and all individuals were subjected to wavelet transformation. Wavelet transform is a critical method due to its capability for time-frequency localization. A key aspect of wavelet analysis is the selection of an appropriate wavelet function. The similarity between the wavelet function and the signals being analyzed is crucial for extracting useful information. In the literature, Morse, Morlet, and Bump wavelets are commonly used. For this study, the Morlet wavelet function was chosen due to its excellent frequency localization properties.

The continuous wavelet transform is defined as the convolution of the original signal $x(t)$ with the wavelet function $\psi_{\tau,s}(t)$, and is represented by Equation 1.

$$SDD_x^{\psi}(\tau, s) = \frac{1}{\sqrt{|s|}} \int x(t)\psi^* \left(\frac{t-\tau}{s}\right) dt$$

(1)

In the equation, $\psi_{(\tau,s)}(t)$ represents the scaled and shifted version of the wavelet function $\psi(t)$, and is defined as shown in Equation 2.

**Table 1. Distribution of segments based on participant responses.**

|  | FF | FR | RF | RR | total |
|---|---|---|---|---|---|
| S01 | 15 | 46 | 17 | 44 | 122 |
| S02 | 27 | 57 | 22 | 50 | 156 |
| S03 | 58 | 50 | 19 | 84 | 211 |
| S04 | 40 | 132 | 11 | 42 | 225 |
| S05 | 60 | 76 | 15 | 57 | 208 |
| S06 | 20 | 58 | 28 | 78 | 184 |
| S07 | 54 | 73 | 10 | 38 | 175 |
| S08 | 32 | 72 | 21 | 62 | 187 |
| S09 | 26 | 81 | 22 | 71 | 200 |
| S10 | 31 | 63 | 12 | 88 | 194 |
| S11 | 41 | 38 | 14 | 57 | 150 |
| S12 | 30 | 59 | 50 | 84 | 223 |
| S13 | 23 | 109 | 15 | 72 | 219 |
| S14 | 30 | 66 | 22 | 84 | 202 |
| S15 | 13 | 55 | 40 | 124 | 232 |
| S16 | 26 | 57 | 14 | 59 | 156 |
| total | 526 | 1092 | 332 | 1094 | 3044 |

$$\psi_{\tau,s}(t) = \frac{1}{\sqrt{s}}\psi\left(\frac{t-\tau}{s}\right)$$

(2)

Here, $t$, $\tau$, and $s$ represent the time, translation, and scaling parameters, respectively. The wavelet function has zero mean, as shown in Equation 3.

$$\int_{-\infty}^{+\infty} \psi_{\tau,s}(t)\, dt = 0$$

(3)

The sum of the squares of the magnitudes of the wavelet transform coefficients (wavelet transform coefficient power) is calculated as shown in Equation 4.

$$WTCP = \sum_{i=1}^{n} \left|w_i\right|^2$$

(4)

In the equation, $w_i$ represents the $i$-th wavelet transform coefficient, and $n$ denotes the total number of wavelet coefficients.

**fNIRS data processing**

In this study, we utilized an existing fNIRS dataset provided in.csv format, which contains raw measurements obtained using the Hitachi ETG-4000 system [27]. To enable further analysis within the Homer2 software environment, we converted the.csv files into the fNIRS format using the hitachi2nirs MATLAB function, available at [30]. This script facilitates the transformation of Hitachi ETG-4000 output files into a format compatible with Homer2 and requires an accompanying. pos file to define the spatial configuration of the optodes. As our dataset did not include digitized optode positions, we adopted one of the example. pos files provided by the developers, corresponding to the standard optode arrangements supported by the system.

Data preprocessing was performed using a combination of nirsLAB (NIRX Medical Technologies, Berlin, Germany), Homer2, and customized scripts developed in MATLAB (Mathworks, Natick, MA, USA). A preprocessing pipeline was constructed based on the functions provided by the Homer2 toolbox [31]. Raw light intensity data were converted into optical density (OD) changes using the hmrIntensity2OD function. Time segments affected by motion artifacts were identified with the hmrMotionArtifact.m function using the parameters: tMotion = 0.5, tMask = 1, STDEVthresh = 10, and AMPthresh = 1. To correct motion artifacts, a spline interpolation method was applied using the hmrMotionCorrectSpline.m function available in the Homer2 toolbox [32]. The motion-corrected OD data were then passed through a third-order Butterworth band-pass filter with high-pass and low-pass cutoff frequencies set at 0.0001 Hz and 0.1 Hz, respectively. Concentrations of oxygenated hemoglobin (HBO) and deoxygenated hemoglobin (HBR) were computed from the filtered OD data using the hmrOD2Conc.m function. Only HBO signals were included in the analysis, as they are considered more reliable indicators of cortical activation [33,34].

fNIRS signals are known to include not only neuronally driven HBO components the primary signals of interest but also systemic physiological components such as cardiac pulsation, respiration, and Mayer waves, originating from both cerebral and extracerebral sources. These physiological effects can introduce false positives or negatives in the resulting hemodynamic activation maps. To address this issue, a principal component analysis approach, as described by Zhang et al. [35], was applied to separate neuronally-induced HBO signals from systemic physiological noise. The top principal components accounting for 90% of the covariance across all channels were treated as regressors representing global physiological noise and subsequently removed from each channel's data.

For each channel, HBO time series data were segmented into 12-second epochs, starting 1 second before and extending 11 seconds after the onset of each stimulus presentation (FF, FR, RF, and RR conditions). For every subject and channel, condition-specific average signals were obtained by computing the mean across all corresponding trials, resulting in a single block-averaged time series per condition and channel. To quantify the amplitude of stimulus-evoked responses, Cohen's d effect size metric was calculated for each condition. Specifically, the mean signal within the 11 second post-stimulus interval was subtracted from the mean of the 1-second pre-stimulus baseline, and the resulting difference was divided by the standard deviation of the same baseline period, consistent with the procedure described in Balconi and Vanutelli [36]. The mathematical formulation of Cohen's d is presented in Equation (5), where $m_{trial}$ denotes the post-stimulus mean, $m_{baseline}$ denotes the pre-stimulus baseline mean, and $s_{baseline}$ represents the baseline standard deviation. To assess statistically significant hemodynamic activations at the group level, one-way ANOVA tests were performed on the Cohen's d values for each channel separately under each stimulus condition. An overview of the full preprocessing

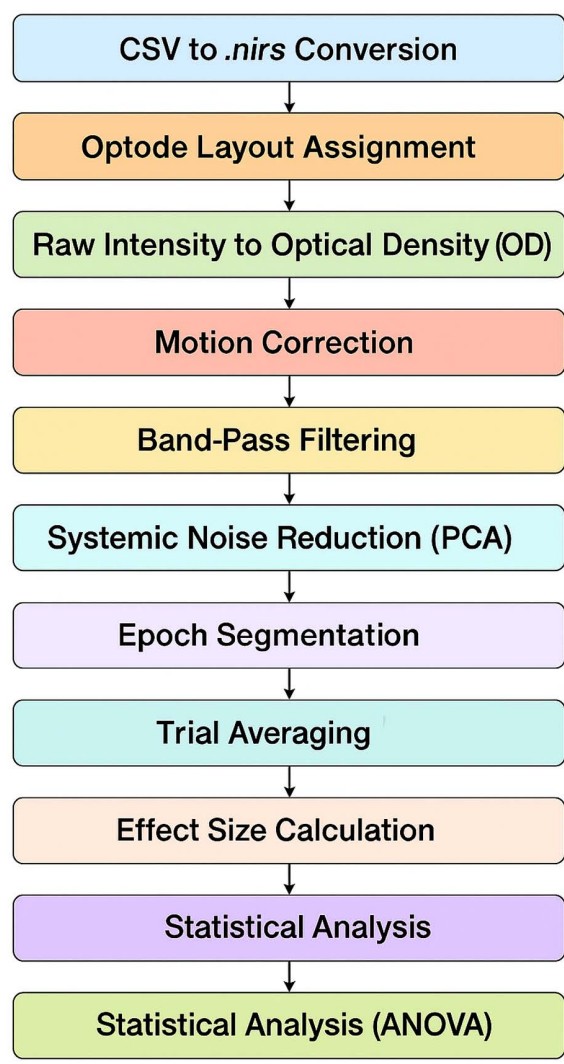

**Fig 3. fNIRS preprocessing and analysis pipeline.**

and analysis workflow applied to the fNIRS signals including data conversion, artifact correction, filtering, segmentation, and statistical modeling is provided in Fig 3.

$$d = \frac{m_{trial} - m_{baseline}}{s_{baseline}}$$

(5)

## Results

The dataset used in this study comprises both EEG and fNIRS recordings acquired simultaneously during a memory-related visual task. EEG analysis focused on the early neural responses elicited during the first second following stimulus presentation, referred to as the "stimulus-locked epoch". Within this window, ERP analyses were conducted on a channel-by-channel basis for four experimental conditions: FF, FR, RF, and RR. To further characterize the temporal and spectral features of these ERP responses, a wavelet transform offering joint time-frequency resolution was applied for each condition. In the subsequent decision-making phase, defined as the 9-second interval following stimulus onset during which participants decided whether or not to remember the image, hemodynamic changes in the fNIRS signals were analyzed. The results section presents the findings from these analyses in the order in which they were conducted.

As shown in Table 1, a total of 526 trials were recorded for the FF stimulus type, 1092 for FR, 332 for RF, and 1094 for RR, based on participants' responses. Since ERP signal quality improves with higher trial counts due to an increased signal-to-noise ratio (SNR), an equal number of trials across conditions was ensured to avoid bias. Specifically, for each stimulus type, 332 trials were randomly selected matching the lowest available count (RF condition) and averaged to compute ERP signals. Fig 4 presents the grand-averaged ERP waveforms obtained from all participants for the following electrode channels: Fz, Cz, P3, P4, P5, P6, P7, P8, O1, O2, PO3, and PO4. In the ERP plots, the x-axis represents time in milliseconds (ms), while the y-axis denotes amplitude in microvolts (µV). The vertical red line marks stimulus onset; the 1000 ms segment to the right of this line reflects the post-stimulus ERP response, whereas the 500 ms segment to the left corresponds to the pre-stimulus baseline period. In the figures, the solid blue line represents the FF condition, the dashed orange line corresponds to FR, the dotted yellow line denotes RF, and the purple dash-dotted line illustrates the RR condition.

Fig 4 presents the grand-averaged ERP responses across a set of frontal, central, parietal, and occipital electrodes, including Fz, Cz, P3, P4, P5, P6, P7, P8, PO3, PO4, O1, and O2, for the four stimulus conditions: RR, RF, FR, and FF. Around 300 ms after stimulus onset, ERP amplitudes associated with remember intention based conditions (RR and RF) tend to be higher than those elicited by others (FR and FF), particularly in parietal and occipital regions. This suggests that stimuli associated with an intention to remember regardless of successful or unsuccessful elicit stronger neural responses in these regions. While this general trend is observed across many electrodes, the most distinct differences are found in channels P5, P6, P8, PO3, PO4, and O1. These findings indicate that the intention to remember enhances early neural processing, as reflected by increased ERP amplitudes in parietal and occipital regions. Additionally, the observed peak around 300 ms post stimulus is suggestive of a P3-like component, which is commonly associated with attentional allocation and memory encoding processes [37,38]. Notably, across all electrodes, the FF condition consistently elicited the lowest ERP amplitudes, which is an expected finding; stimuli that were both not intended to be remembered and were ultimately forgotten appear to evoke minimal neural engagement during early cognitive processing.

To gain a more detailed understanding of the temporal and spectral characteristics of the ERP signals, time-frequency analysis was conducted using wavelet transform. This method allows simultaneous examination of neural responses across both time and frequency domains. The analysis focused on ten electrodes located in the parietal and occipital regions P3, P4, P5, P6, P7, P8, PO3, PO4, O1, and O2 where ERP amplitude differences were most prominent in the previous analyses. For each of these electrodes, continuous wavelet transforms were applied to the averaged ERP signals. The resulting time-frequency representations are presented in Fig 5. In these plots, the x-axis represents time in

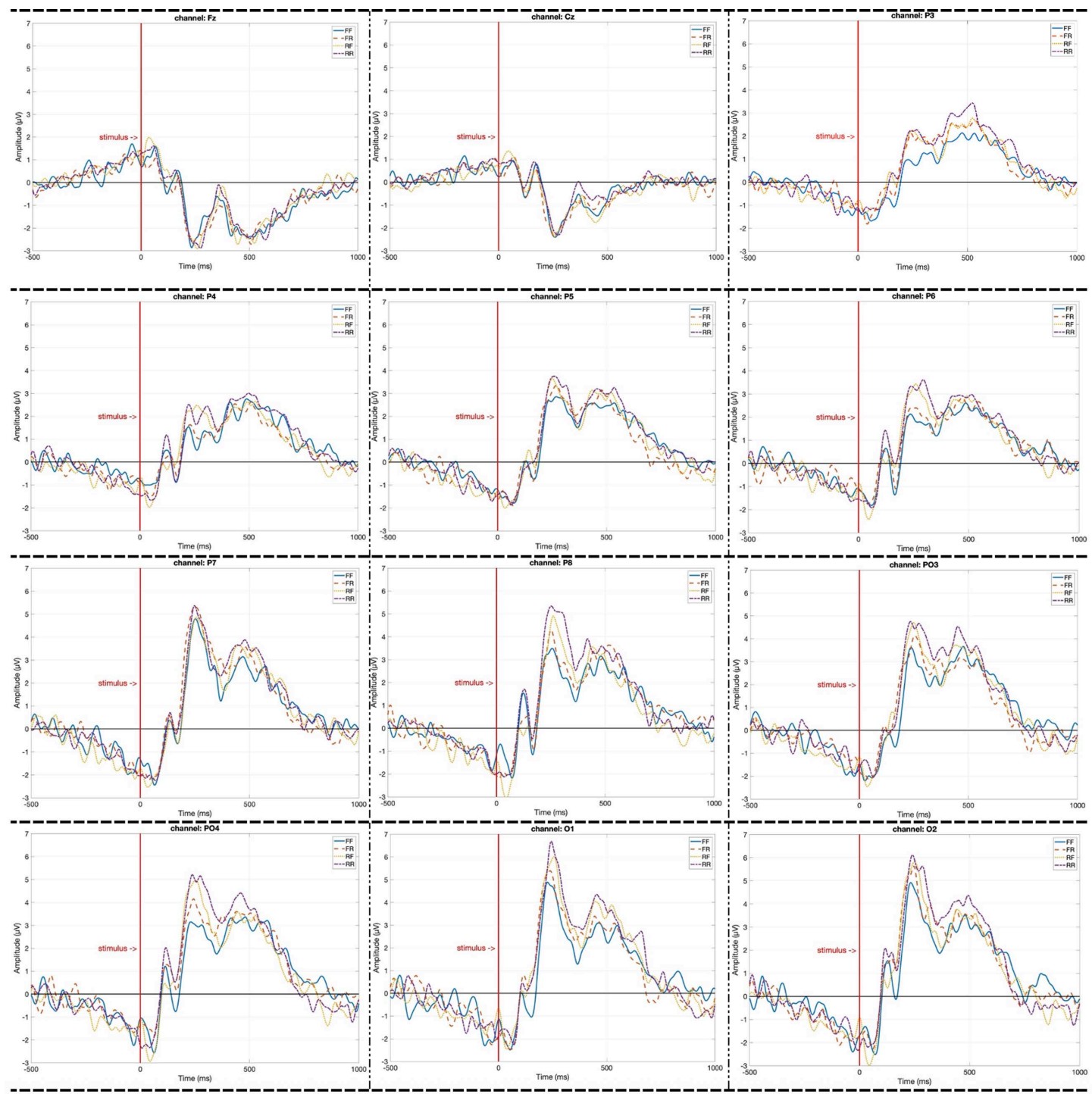

**Fig 4. Grand-averaged ERP waveforms across channels for each stimulus condition.**

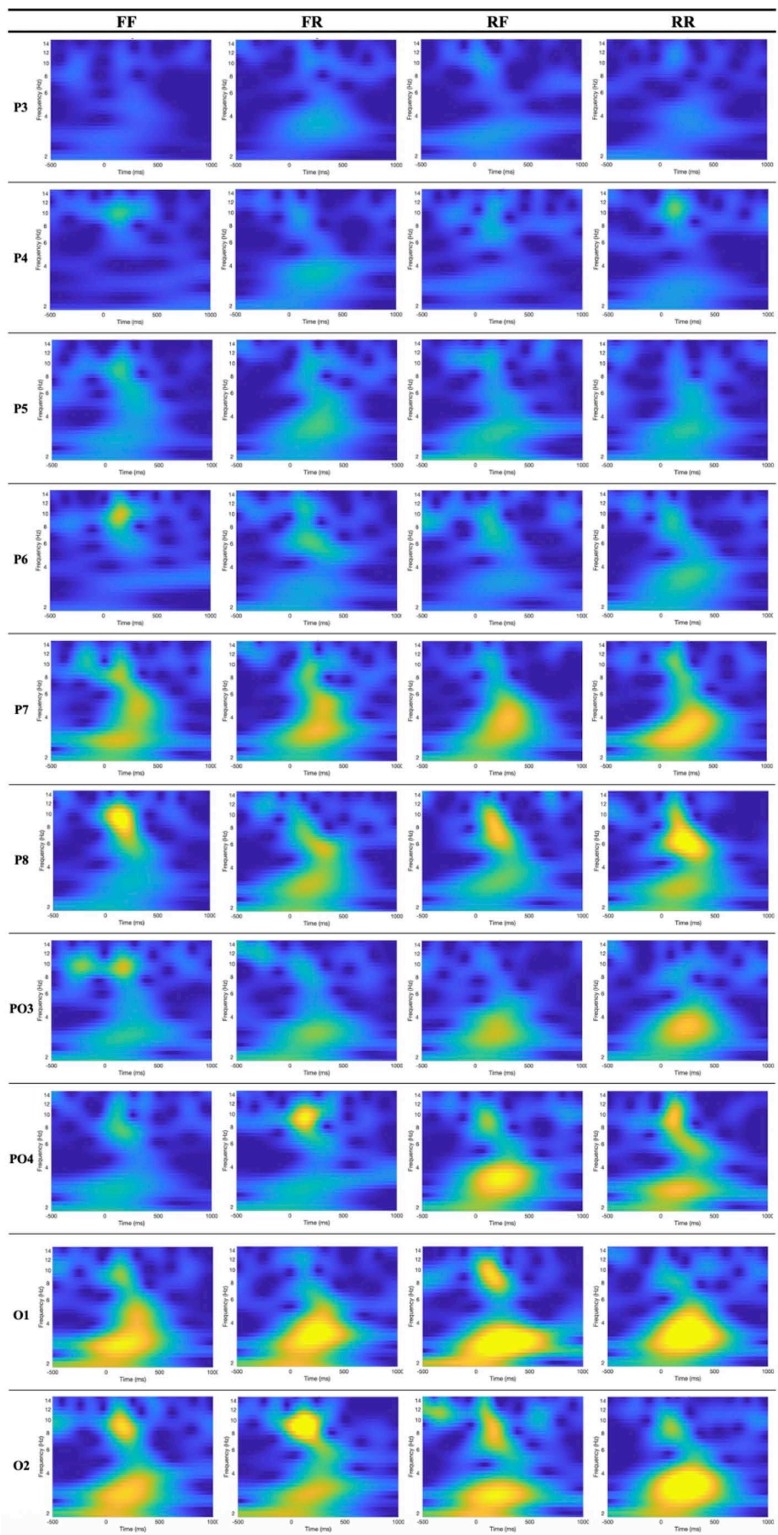

**Fig 5. Time-frequency representations of ERP signals using wavelet transform.**

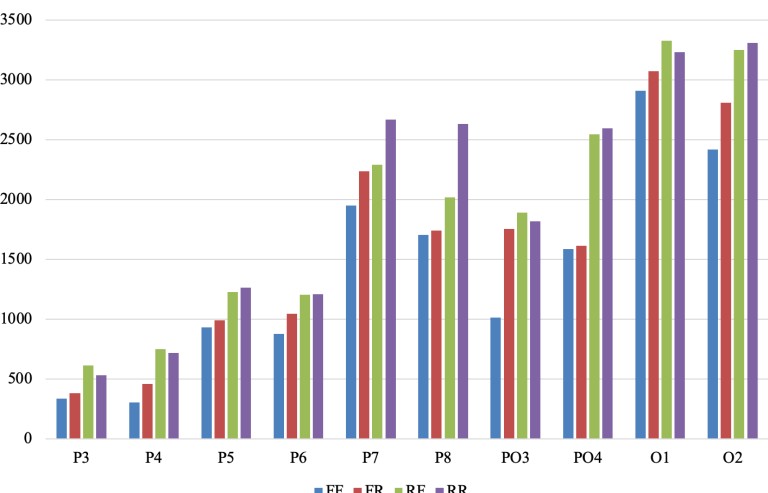

**Fig 6. Total wavelet power coefficients across parietal and occipital electrodes for each stimulus condition.**

milliseconds, and the y-axis shows frequency components ranging from 1 to 15 Hz. These visualizations provide insights into the dynamic frequency content of the ERP responses across different stimulus conditions and help to characterize the underlying neural mechanisms associated with memory intention and performance.

These visualizations display the temporal evolution of frequency components (1–15 Hz) following stimulus onset across the four experimental conditions: FF, FR, RF, and RR. Consistent with the ERP findings, the wavelet power patterns indicate greater activity in the RR and RF conditions, particularly within the lower frequency range, which is characteristic of theta and low alpha bands. This enhanced activity is most prominent in channels O1, O2, PO3, PO4, P7, and P8, as reflected by larger and more sustained yellow regions in the time-frequency plots. As a representative example, channel PO4 displayed enhanced wavelet power in the theta and low alpha frequency bands, particularly under the RR and RF conditions, supporting the overall trend of stronger neural engagement during intentional memory processing. These results suggest that the intention to remember not only modulates ERP amplitude in the time domain but also strengthens neural oscillatory activity in the time-frequency domain, especially in parietal and occipital regions. The increased power in RR and RF conditions is predominantly observed within the 0–500 ms post-stimulus interval.

To further quantify the findings from the time-frequency representations, wavelet transform coefficients were extracted for each condition and electrode, as described in the Materials and Methods section. Fig 6 presents the total wavelet power values for the ten parietal and occipital electrodes across the four stimulus conditions. These numerical results support the qualitative observations made from the time-frequency plots: higher wavelet coefficients are consistently observed for the RR and RF conditions compared to FR and FF. The strongest wavelet power values were observed in O1, O2, PO4, P7, and P8 channels. For instance, in the O1 channel, RR and RF conditions yielded coefficients of 3232.83 and 3324.29 respectively, which are substantially higher than those for FR (3073.21) and FF (2908.15). A similar trend is evident across other electrodes. These findings align with ERP-based results, highlighting that memory intention not only modulates time-domain amplitude responses but also enhances time-frequency energy, especially in occipital and parietal regions.

To investigate the hemodynamic correlates of memory-related decision making, analyzed fNIRS signals during interval following stimulus presentation, corresponding to the period in which participants decided whether or not they wanted to remember the image. This decision-making phase was examined across all four stimulus conditions. Hemodynamic responses were visualized using block-averaged HbO signals for each condition to capture the cortical dynamics

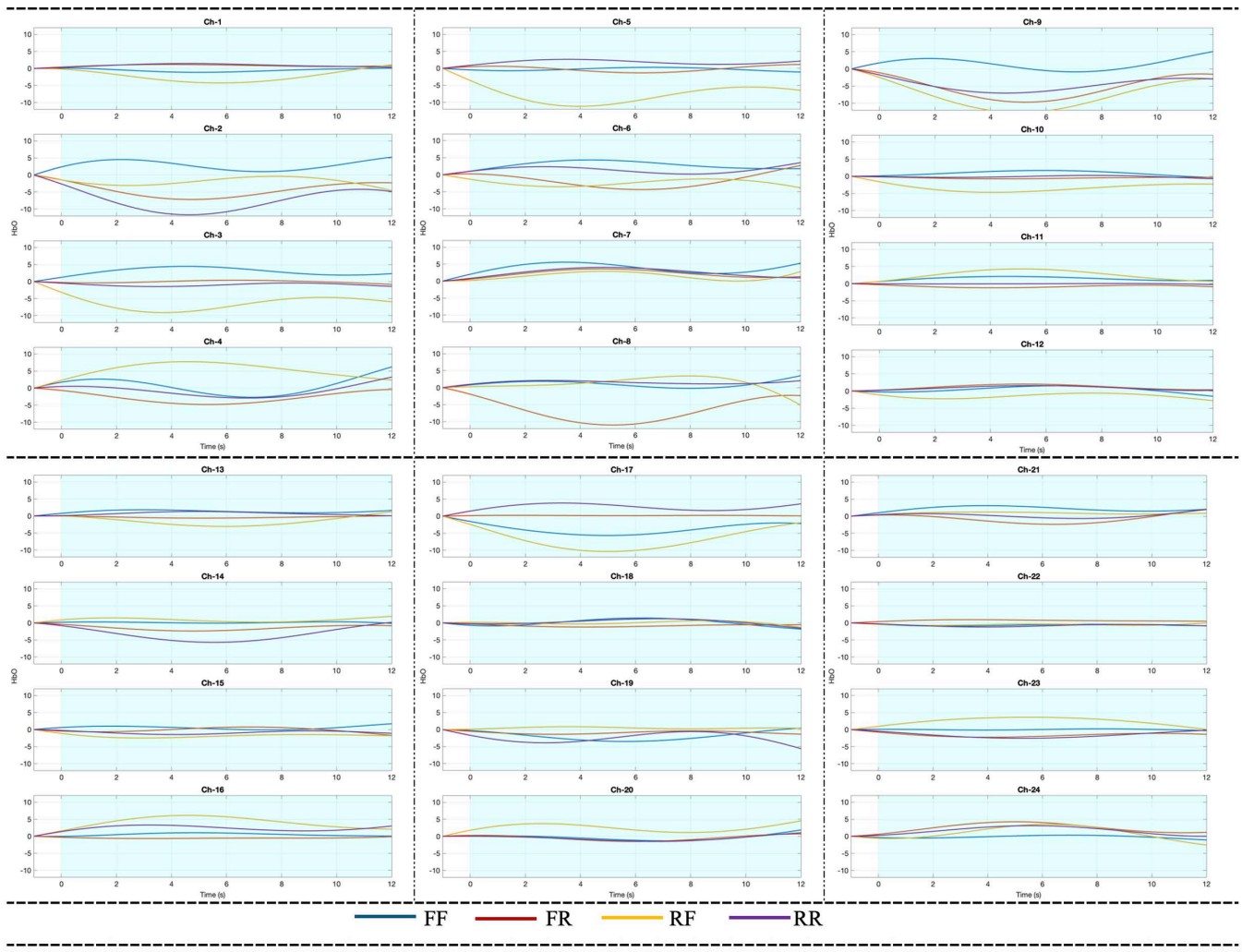

**Fig 7. Hemodynamic responses during decision Phase.**

associated with decision performance. The resulting activation patterns are presented in Fig 7, based on signals recorded from 24 fNIRS channels distributed across the frontal, temporal, parietal, and occipital regions. In the visualizations, the x-axis represents time in seconds, while the y-axis indicates the concentration of HbO in millimoles per milliliter (mMol/ml). The time point 0.ms marks the onset of the stimulus. Each plot displays a 12-second time window: the 1-second interval preceding stimulus onset serves as the pre-stimulus baseline, while the 11 seconds following the stimulus capture the subsequent hemodynamic response. Of this post-stimulus period, the first 9 seconds correspond to the decision-making phase, and the remaining 2 seconds illustrate how the hemodynamic signal evolves immediately after the decision has been made.

Fig 7 illustrates the block-averaged HbO responses across the four stimulus conditions. Similar patterns were observed across all 24 channels. While there are observable variations in HbO amplitude between conditions, no consistent or systematic trend emerges that clearly differentiates the stimulus types across the entire cortex. In some channels, higher responses are seen for FF, while in others RF, FR, or RR dominate, suggesting that the hemodynamic response during the decision-making phase may vary depending on cortical region and individual-level factors. In addition to the graphical

summary, descriptive statistics for each condition were computed to enhance the interpretability and transparency of the hemodynamic signal distribution. Table 2 presents the mean±standard deviation values of HbO concentrations during the decision phase across all 24 channels and four stimulus conditions. This tabulated summary complements the visualization by providing quantitative insight into intra-condition variability and between-condition comparisons.

To enable a more robust and interpretable comparison between conditions, Cohen's D effect size values were calculated for each channel and condition, as detailed in the Materials and Methods section. These values were then used to conduct statistical analyses, the results of which are presented in the following section. Fig 8 displays the distribution of Cohen's D effect size values for each of the 24 fNIRS channels across the four stimulus conditions (FF, FR, RF, RR). The bar plots represent the magnitude of condition-specific hemodynamic changes during the decision-making phase. While certain channels (e.g., Ch13, Ch15, Ch17, Ch23) exhibit relatively higher effect sizes for FF condition, the overall pattern remains heterogeneous, with no single stimulus condition consistently producing stronger activation across all channels. To statistically evaluate whether these differences were significant, one-way ANOVA tests were performed for each channel, comparing the Cohen's D values across the four conditions. The results revealed that none of the channels exhibited statistically significant differences between conditions ($p > 0.05$). These findings suggest that, although variations in hemodynamic response magnitude exist, they do not reach a level of significance to indicate a consistent or condition-specific effect at the group level.

**Table 2. Channel-wise descriptive statistics of HbO responses.**

| Channel | FF | FR | RF | RR |
|---|---|---|---|---|
| 1 | −0.46±2.5 | 0.79±2.8 | −2.1±2.6 | 0.87±3.0 |
| 2 | 2.78±5.6 | −4.34±5.1 | −1.82±6.1 | −7.2±5.3 |
| 3 | 2.86±3.7 | −0.07±3.8 | −6.12±5.7 | −0.9±4.4 |
| 4 | 0.61±4.9 | −2.72±4.8 | 5.18±5.1 | −0.79±4.7 |
| 5 | −0.28±4.9 | −0.14±5.2 | −7.46±5.3 | 1.72±4.9 |
| 6 | 2.73±4.3 | −1.71±4.5 | −2.26±5.1 | 1.38±4.6 |
| 7 | 3.65±4.7 | 2.09±5.7 | 1.39±4.9 | 2.47±5.3 |
| 8 | 1.02±6.5 | −6.41±6.5 | 1.2±7.7 | 1.48±2.3 |
| 9 | 1.42±4.9 | −5.25±4.6 | −7.58±4.8 | −4.46±5.7 |
| 10 | 0.85±2.9 | −0.47±3.0 | −3.21±4.0 | −0.12±4.8 |
| 11 | 1.38±2.3 | −0.76±2.5 | 2.45±2.0 | −0.05±3.4 |
| 12 | 0.41±2.9 | 1.1±3.0 | −1.43±2.9 | 0.84±2.7 |
| 13 | 1.26±2.1 | −0.31±2.5 | −1.41±2.2 | 0.7±3.0 |
| 14 | 0.14±3.3 | −1.38±3.0 | 0.88±3.5 | −3.2±3.3 |
| 15 | 0.55±2.8 | −0.11±2.9 | −1.76±3.1 | −0.81±4.0 |
| 16 | 0.48±3.7 | −0.49±4.1 | 3.93±3.9 | 2.25±4.5 |
| 17 | −3.65±6.1 | 0.19±3.6 | −6.56±5.8 | 2.56±3.8 |
| 18 | 0.07±3.1 | −0.78±2.3 | 0.02±3.1 | 0.25±4.1 |
| 19 | −1.82±4.1 | −0.86±2.8 | 0.43±2.7 | −2.35±2.4 |
| 20 | −0.32±3.2 | −0.43±2.7 | 2.41±3.8 | −0.54±3.2 |
| 21 | 2.07±2.6 | −0.71±3.4 | 0.86±1.2 | 0.23±1.2 |
| 22 | −0.62±2.9 | 0.68±2.0 | −0.49±2.3 | −0.71±1.9 |
| 23 | 0.02±2.2 | −1.51±2.5 | 2.38±3.1 | −1.52±2.3 |
| 24 | −0.18±3.3 | 2.38±1.4 | 0.99±2.8 | 1.6±4.3 |

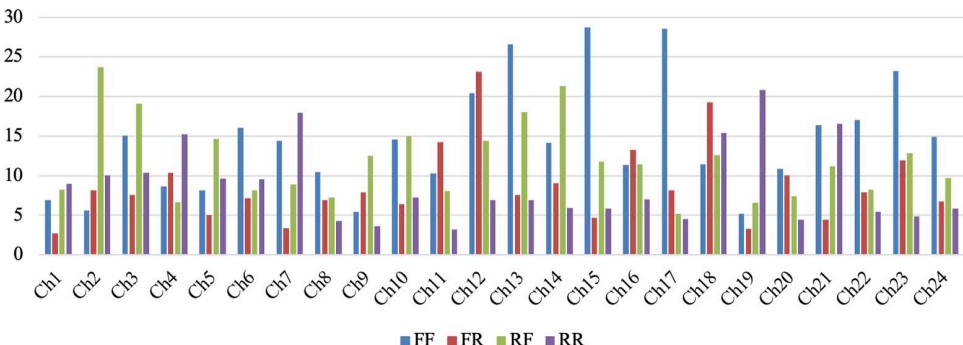

**Fig 8. Cohen's D effect size values across 24 fNIRS channels for each stimulus condition.**

## Conclusion

This study investigated the neural and hemodynamic correlates of intentional memory processing by leveraging a multi-modal dataset comprising simultaneous EEG and fNIRS recordings collected during a visual memory task. Participants viewed images and were later asked to decide whether they wanted to remember or forget each stimulus. Four experimental conditions were defined based on the interaction between intention and actual memory performance: RR, RF, FR, and FF.

ERP analyses focusing on the first second following stimulus presentation revealed that the RR and RF conditions both associated with the intention to remember elicited stronger neural responses compared to FR and FF, particularly in parietal and occipital regions. These differences emerged around 300 ms post-stimulus and were most pronounced in electrodes such as P5, P6, P8, PO3, PO4, and O1. This temporal window aligns with the P3 component, which is typically linked to attentional resource allocation and memory encoding processes. The consistent finding that FF stimuli evoked the weakest ERP amplitudes further supports the idea that lack of intentional engagement results in reduced early cortical processing. To complement the ERP findings and provide a more nuanced view of underlying neural dynamics, time-frequency analysis using wavelet transforms was performed. This approach allowed simultaneous inspection of the temporal and spectral properties of the ERP signals. The results again highlighted greater power in the theta and low alpha bands for RR and RF conditions, particularly in posterior channels such as O1, O2, PO3, PO4, P7, and P8. These low-frequency enhancements, especially during the early post-stimulus phase (0–500 ms), suggest that intentional memory engagement modulates not only amplitude but also oscillatory power, reflecting increased cognitive effort and encoding efficiency.

In the subsequent decision-making phase captured by 9-second post-stimulus window hemodynamic responses were assessed using fNIRS signals. The HbO signals did not exhibit a consistent condition-specific pattern across the 24 recorded channels, which spanned frontal, temporal, parietal, and occipital regions. Although variations in response magnitude were observed across conditions and channels, no clear or systematic trend emerged. This variability was further investigated through effect size analysis using Cohen's D values, which were computed for each channel and stimulus condition. However, statistical testing using one-way ANOVA revealed no significant differences between conditions at any channel ($p > 0.05$), suggesting that the hemodynamic responses during the decision-making period may reflect more distributed or individually variable patterns of brain activity.

In summary, the findings demonstrate that the intention to remember significantly influences early electrophysiological responses, as captured by ERP amplitude and time-frequency dynamics. These effects are most robust in parietal and occipital regions and are specifically enhanced for stimuli that participants attempted to remember. In contrast, the corresponding hemodynamic signals during the decision-making phase, while informative, did not show consistent condition-related modulations at the group level. Factors such as inter-individual variability in hemodynamic responses, low

signal-to-noise ratio inherent to fNIRS measurements, and the relatively limited spatial resolution of fNIRS systems may have contributed to the absence of significant differences across conditions. This discrepancy highlights the complementary nature of EEG and fNIRS: EEG captures rapid, transient neural responses closely linked to cognitive processes such as attention and encoding, whereas fNIRS reflects slower, cumulative metabolic changes that may be more susceptible to interindividual differences and task complexity. Overall, this study underscores the value of combining EEG and fNIRS to investigate the multidimensional nature of memory processing. While ERP and time-frequency findings clearly support the role of intentional engagement in enhancing memory-related brain activity, further work is needed to better characterize how these processes manifest in hemodynamic signals especially during more complex cognitive phases such as decision making.

## Author contributions

**Formal analysis:** Tolga Turay.

**Investigation:** Tolga Turay, Onur Erdem Korkmaz.

**Methodology:** Tolga Turay, Onur Erdem Korkmaz.

**Project administration:** Tolga Turay.

**Software:** Onur Erdem Korkmaz.

**Supervision:** Onur Erdem Korkmaz.

**Validation:** Tolga Turay, Onur Erdem Korkmaz, Ebru Ergün.

**Visualization:** Onur Erdem Korkmaz.

**Writing – original draft:** Tolga Turay, Onur Erdem Korkmaz, Ebru Ergün.

**Writing – review & editing:** Onur Erdem Korkmaz, Ebru Ergün.

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
