## [Decision Letter · Decision Letter 0]

28 Apr 2025

Dear Dr. ERGÜN,

Thank you for submitting your manuscript to PLOS ONE. After careful consideration, we feel that it has merit but does not fully meet PLOS ONE’s publication criteria as it currently stands. Therefore, we invite you to submit a revised version of the manuscript that addresses the points raised during the review process.

We look forward to receiving your revised manuscript.

Kind regards,

Prof. Dr. Önder Aydemir

Academic Editor

PLOS ONE

 [This research was financially supported by the Recep Tayyip Erdogan University Development Foundation. We sincerely appreciate their support, which contributed significantly to the completion of this study.]. 

[This research was financially supported by the Recep Tayyip Erdogan University Development Foundation (Grant number: 02025004007391). We sincerely appreciate their support, which contributed significantly to the completion of this study.]

 [This research was financially supported by the Recep Tayyip Erdogan University Development Foundation. We sincerely appreciate their support, which contributed significantly to the completion of this study.]. 

Reviewers' comments:

Reviewer's Responses to Questions

**Comments to the Author**

1. Is the manuscript technically sound, and do the data support the conclusions?

Reviewer #1: Yes

Reviewer #2: Yes

2. Has the statistical analysis been performed appropriately and rigorously?

Reviewer #1: Yes

Reviewer #2: Yes

3. Have the authors made all data underlying the findings in their manuscript fully available?

Reviewer #1: Yes

Reviewer #2: Yes

4. Is the manuscript presented in an intelligible fashion and written in standard English?

Reviewer #1: Yes

Reviewer #2: Yes

Reviewer #1: General Evaluation: This manuscript presents a well-structured multimodal neuroimaging study combining EEG and fNIRS signals to examine motivation-related visual cognitive processes. The topic is timely, and the methodology is clearly described. However, the manuscript would benefit from several improvements regarding figure clarity, data reporting, and statistical transparency.

General Minor Comments

• The abstract is mostly clear; however, the sentence beginning with "Although descriptive variability in HbO..." should be rephrased for better clarity and flow.

• A diagram or flowchart summarizing the fNIRS preprocessing pipeline would significantly improve clarity for readers unfamiliar with the multi-step procedure.

• In the Results section, Tables 3 and 4 could benefit from deeper commentary. Consider walking through a representative channel (e.g., PO4) in more detail.

• In the Discussion, the non-significance of fNIRS results should be explored more thoroughly. Possible explanations may include inter-individual variability, low signal-to-noise ratio, or spatial resolution limitations.

• There are some stylistic redundancies throughout the manuscript (e.g., repeated definitions of “motivation” and “ERP”). These could be condensed to enhance readability.

• Some sentences are overly long or passive in structure. Refining the language for clarity and conciseness would improve the paper’s flow.

Major Comments

1. Clarification of EEG and fNIRS Channel Locations (Figure 2) In Figure 2, EEG and fNIRS sensors are plotted simultaneously; however, it remains unclear which markers correspond to EEG electrodes and which to fNIRS optodes. Please clearly distinguish them either through labels, shapes, or a color legend. I also recommend updating the figure caption to explicitly indicate which symbols represent EEG and fNIRS channels.

Suggested caption addition: "EEG electrodes are shown as [color/symbol], whereas fNIRS optodes (sources and detectors) are shown as [color/symbol]."

2. Missing Formula for Cohen’s d (Figure 4) Cohen’s d is used to generate Figure 4, but the manuscript does not state the formula used for its computation. Please include the formula or a clear citation to ensure reproducibility.

3. Inconsistent Y-Axis Ranges in Figure 10 The subplots in Figure 10 appear to use varying y-axis scales, which hinders visual comparison between conditions. I recommend standardizing the y-axis (e.g., from –10 to 10) across all plots unless variability is justified and explained in the caption.

Suggested note: "All y-axes were scaled uniformly from –10 to 10 µV for consistent comparison."

4. Missing Descriptive Statistics for Hemodynamic Responses (Table 4) Table 4 summarizes fNIRS responses but lacks descriptive statistics such as means and variances. Including a supplementary table that reports mean ± standard deviation values would enhance transparency and allow for deeper interpretation of the hemodynamic data.

Suggested improvement: "A summary table reporting the mean and variance of HbO/HbR values per condition would strengthen the results."

5. Inconsistent Image Ordering in Table 4 In Table 4, the order of images across rows is inconsistent, which may confuse readers. Ensure that the same ordering of visual elements (conditions, time points, or regions) is maintained across all rows for clarity.

Suggested instruction: "Use a fixed left-to-right ordering of conditions across all rows (e.g., [A, B, C])."

Reviewer #2: The manuscript presents a well-organized and timely multimodal study using EEG and fNIRS to investigate visual cognitive motivation. The topic is important and the methodological framework is overall sound. However, significant revisions are necessary before the manuscript can be considered for publication. The main points requiring attention are outlined below:

The simultaneous plotting of EEG electrodes and fNIRS optodes in Figure 2 is currently unclear. It is difficult for the reader to distinguish which symbols correspond to EEG and which to fNIRS components. Please revise the figure by using different shapes and/or colors for EEG electrodes and fNIRS sources/detectors, and update the figure caption accordingly. This distinction is crucial for understanding the multimodal setup. While the fNIRS preprocessing pipeline is described in detail, readers unfamiliar with fNIRS analysis might find it difficult to follow the multiple steps. Including a flowchart or diagram summarizing the preprocessing steps (motion correction, filtering, PCA, epoching, etc.) would substantially enhance the clarity and accessibility of this section. Cohen’s d values were used extensively, especially in interpreting fNIRS results (e.g., Figure 4), but the exact formula for Cohen’s d calculation is not provided. Please explicitly state the formula or provide a precise reference. Reproducibility is a key requirement. Subplots in Figure 10 (fNIRS responses) have inconsistent y-axis ranges across conditions. This inconsistency complicates visual comparison. Standardizing the y-axis scale across all subplots (e.g., –10 to 10 µV for EEG plots, appropriate range for HbO concentration in fNIRS) would improve interpretability. If variability requires separate scaling, this should be justified clearly in the figure captions. Table 4 presents averaged fNIRS hemodynamic responses but lacks descriptive statistics (mean ± standard deviation) across participants. This omission reduces the transparency and interpretability of the findings. Please add a supplementary table including means and variances for each condition and channel.

**Do you want your identity to be public for this peer review?** For information about this choice, including consent withdrawal, please see our Privacy Policy

Reviewer #1: No

Reviewer #2: No

---

## [Author Response · Author response to Decision Letter 1]

1 May 2025

AUTHOR RESPONSES TO THE REVIEWERS

Journal: PLOS ONE

Title of the manuscript: Simultaneous EEG-fNIRS Study of Visual Cognitive Processing: ERP Analysis and Decision-Related Hemodynamic Responses in Healthy Adults

Manuscript ID Number: PONE-D-25-18575

Authors: Tolga Turay, Onur Erdem Korkmaz, Ebru Ergün

We would like to start by thanking the editor and the reviewer for their useful comments and suggestions on how to further improve the quality of our manuscript. We have been able to incorporate changes to reflect all the suggestions provided by the reviewer. Here is a point-by-point response to the reviewer’s comments and concerns.

Reviewer 1:

1. Asked: “The abstract is mostly clear; however, the sentence beginning with "Although descriptive variability in HbO..." should be rephrased for better clarity and flow.”

1. Answer: We thank the reviewer for this valuable suggestion. In response, we have revised the sentence to improve clarity and readability. The updated version now reads: “While visual inspection revealed variability in oxygenated hemoglobin (HbO) levels across channels and conditions, statistical analyses using Cohen’s D and one-way ANOVA did not identify any significant differences between the conditions (p > 0.05).”

2. Asked: “A diagram or flowchart summarizing the fNIRS preprocessing pipeline would significantly improve clarity for readers unfamiliar with the multi-step procedure”

2. Answer: We appreciate the reviewer’s insightful suggestion to improve the clarity of our methodology. In response, we have added a new visual summary illustrating the main steps of the fNIRS preprocessing pipeline. This diagram is now included as Figure 3 in the revised manuscript and provides an overview of the key stages from raw data conversion to statistical analysis.

3. Asked: “In the Results section, Tables 3 and 4 could benefit from deeper commentary. Consider walking through a representative channel (e.g., PO4) in more detail.”

3. Answer: We thank the reviewer for this valuable suggestion. In response, we added a specific commentary highlighting the PO4 channel after the description of Table 3. This additional note emphasizes the enhanced wavelet power observed in the theta and low alpha bands for the RR and RF conditions, providing a representative example of the observed patterns.

4. Asked: “In the Discussion, the non-significance of fNIRS results should be explored more thoroughly. Possible explanations may include inter-individual variability, low signal-to-noise ratio, or spatial resolution limitations.”

4. Answer: We thank the reviewer for this important suggestion. In response, we have expanded the Conclusion section by adding a sentence that addresses potential reasons for the lack of significant fNIRS findings, including inter-individual variability, low signal-to-noise ratio, and spatial resolution limitations. This addition follows the initial interpretation of the non-significant HbO results.

5. Asked: “There are some stylistic redundancies throughout the manuscript (e.g., repeated definitions of “motivation” and “ERP”). These could be condensed to enhance readability.”

5. Answer: We thank the reviewer for this valuable observation. In response, the manuscript has been carefully reviewed and revised to eliminate stylistic redundancies, including repeated definitions of key concepts such as “motivation” and “ERP,” thereby improving overall readability and flow.

6. Asked: “Some sentences are overly long or passive in structure. Refining the language for clarity and conciseness would improve the paper’s flow.”

6. Answer: We thank the reviewer for this constructive feedback. Accordingly, the manuscript has been thoroughly revised to improve clarity and conciseness by refining overly long or passive sentences, thereby enhancing the overall flow of the text.

7. Asked: “Clarification of EEG and fNIRS Channel Locations (Figure 2) In Figure 2, EEG and fNIRS sensors are plotted simultaneously; however, it remains unclear which markers correspond to EEG electrodes and which to fNIRS optodes. Please clearly distinguish them either through labels, shapes, or a color legend. I also recommend updating the figure caption to explicitly indicate which symbols represent EEG and fNIRS channels. Suggested caption addition: "EEG electrodes are shown as [color/symbol], whereas fNIRS optodes (sources and detectors) are shown as [color/symbol].”

7. Answer: We thank the reviewer for this important observation. In response, we have updated Figure 2 by adding a clear legend and labels to distinguish EEG electrodes from fNIRS sources and detectors. The figure caption has also been revised to explicitly describe the markers representing EEG and fNIRS components.

8. Asked: “Missing Formula for Cohen’s d (Figure 4) Cohen’s d is used to generate Figure 4, but the manuscript does not state the formula used for its computation. Please include the formula or a clear citation to ensure reproducibility.”

8. Answer: We thank the reviewer for this valuable suggestion. In response, we have added the explicit formula used for the computation of Cohen’s d in the revised manuscript to enhance clarity and reproducibility. A detailed explanation of the parameters involved in the formula has also been provided.

9. Asked: “Inconsistent Y-Axis Ranges in Table 4 The subplots in Table 4 appear to use varying y-axis scales, which hinders visual comparison between conditions. I recommend standardizing the y-axis (e.g., from –10 to 10) across all plots unless variability is justified and explained in the caption. Suggested note: "All y-axes were scaled uniformly from –10 to 10 µV for consistent comparison.”

9. Answer: We thank the reviewer for this important suggestion. In response, Table 4 has been revised, and the y-axis ranges across all subplots have been standardized to allow for consistent and meaningful comparisons between conditions. This update has substantially improved the clarity of the figure.

10. Asked: “Missing Descriptive Statistics for Hemodynamic Responses (Table 4) Table 4 summarizes fNIRS responses but lacks descriptive statistics such as means and variances. Including a supplementary table that reports mean ± standard deviation values would enhance transparency and allow for deeper interpretation of the hemodynamic data. Suggested improvement: "A summary table reporting the mean and variance of HbO/HbR values per condition would strengthen the results.”

10. Answer: We thank the reviewer for this valuable suggestion. In response, a new supplementary table (Table 5) has been created, presenting the mean ± standard deviation values for each channel and stimulus condition. This addition enhances the transparency of the fNIRS results and facilitates deeper interpretation of the hemodynamic patterns.

11. Asked: “Inconsistent Image Ordering in Table 3 In Table 3, the order of images across rows is inconsistent, which may confuse readers. Ensure that the same ordering of visual elements (conditions, time points, or regions) is maintained across all rows for clarity.

Suggested instruction: "Use a fixed left-to-right ordering of conditions across all rows (e.g., [A, B, C]).”

11. Answer: We thank the reviewer for this helpful observation. In response, the inconsistency in the ordering of images in Table 3 has been corrected, and a revised version with a fixed and uniform order across all rows has been included in the manuscript to improve clarity and coherence.

Reviewer 2:

1. Asked: “The simultaneous plotting of EEG electrodes and fNIRS optodes in Figure 2 is currently unclear. It is difficult for the reader to distinguish which symbols correspond to EEG and which to fNIRS components. Please revise the figure by using different shapes and/or colors for EEG electrodes and fNIRS sources/detectors, and update the figure caption accordingly. This distinction is crucial for understanding the multimodal setup.”

1. Answer: We appreciate the reviewer’s observation regarding the clarity of sensor representation in Figure 2. In response, the figure has been revised to visually differentiate EEG electrodes from fNIRS sources and detectors using distinct colors and marker shapes. The figure caption has also been updated to explicitly define each symbol, ensuring that the multimodal setup is easily interpretable.

2. Asked: “While the fNIRS preprocessing pipeline is described in detail, readers unfamiliar with fNIRS analysis might find it difficult to follow the multiple steps. Including a flowchart or diagram summarizing the preprocessing steps (motion correction, filtering, PCA, epoching, etc.) would substantially enhance the clarity and accessibility of this section.”

2. Answer: We thank the reviewer for highlighting the need to improve the accessibility of the preprocessing description. To address this, we have added a flowchart (Figure 3) that visually outlines the major steps of the fNIRS preprocessing pipeline. This figure is intended to assist readers—particularly those less familiar with fNIRS methodology in understanding the sequential structure of the analysis process.

3. Asked: “Cohen’s d values were used extensively, especially in interpreting fNIRS results (e.g., Figure 4), but the exact formula for Cohen’s d calculation is not provided. Please explicitly state the formula or provide a precise reference. Reproducibility is a key requirement.”

3. Answer: We thank the reviewer for emphasizing the importance of transparency in reporting effect size metrics. In response, we have explicitly included the formula used to compute Cohen’s d in the revised manuscript, along with a clear explanation of the corresponding variables. This addition aims to ensure full reproducibility and clarity of the analysis.

4. Asked: “Subplots in Table 4 (fNIRS responses) have inconsistent y-axis ranges across conditions. This inconsistency complicates visual comparison. Standardizing the y-axis scale across all subplots (e.g., –10 to 10 µV for EEG plots, appropriate range for HbO concentration in fNIRS) would improve interpretability. If variability requires separate scaling, this should be justified clearly in the figure captions.”

4. Answer: We appreciate the reviewer’s helpful feedback regarding the visual consistency of Table 4. To address this issue, we have standardized the y-axis ranges across all subplots to facilitate clearer comparisons between conditions. This adjustment improves the interpretability of the data and ensures a more coherent visual presentation.

5. Asked: “Table 4 presents averaged fNIRS hemodynamic responses but lacks descriptive statistics (mean ± standard deviation) across participants. This omission reduces the transparency and interpretability of the findings. Please add a supplementary table including means and variances for each condition and channel.”

5. Answer: We are grateful to the reviewer for pointing out this important omission. To improve the transparency and interpretability of our results, we have created an additional table (Table 5) that presents the mean and standard deviation values of HbO responses for each condition and channel. This supplementary table complements Table 4 and provides a more detailed quantitative summary of the hemodynamic data.

---

## [Decision Letter · Decision Letter 1]

5 May 2025

Simultaneous EEG-fNIRS Study of Visual Cognitive Processing: ERP Analysis and Decision-Related Hemodynamic Responses in Healthy Adults

PONE-D-25-18575R1

Dear Dr. Ebru Ergün,

We’re pleased to inform you that your manuscript has been judged scientifically suitable for publication and will be formally accepted for publication once it meets all outstanding technical requirements.

Kind regards,

Onder Aydemir

Academic Editor

PLOS ONE

Additional Editor Comments (optional):

Thank you for your thorough and thoughtful revisions. The reviewers have confirmed that all their comments were adequately addressed, and the revised manuscript demonstrates improved clarity and rigor. I am pleased to recommend the manuscript for publication. Congratulations on this fine work.

Reviewers' comments:

Reviewer's Responses to Questions

**Comments to the Author**

Reviewer #1: All comments have been addressed

Reviewer #2: All comments have been addressed

2. Is the manuscript technically sound, and do the data support the conclusions?

Reviewer #1: Yes

Reviewer #2: Yes

3. Has the statistical analysis been performed appropriately and rigorously?

Reviewer #1: Yes

Reviewer #2: Yes

4. Have the authors made all data underlying the findings in their manuscript fully available?

Reviewer #1: Yes

Reviewer #2: Yes

5. Is the manuscript presented in an intelligible fashion and written in standard English?

Reviewer #1: Yes

Reviewer #2: Yes

Reviewer #1: Thank you for the revised manuscript. The authors have addressed all of my previous comments satisfactorily. I have no further concerns and recomandation

Reviewer #2: I am pleased to report that the authors have thoroughly addressed all the concerns raised during the initial review. The revisions have significantly improved the clarity and overall quality of the manuscript. All requested changes appear to have been made appropriately, and no further modifications are necessary. Therefore, I recommend that the manuscript be accepted for publication in its current form.

Best regards,

Dr. Tugba Aydemir

**Do you want your identity to be public for this peer review?** For information about this choice, including consent withdrawal, please see our Privacy Policy

Reviewer #1: No

Reviewer #2: No

---

## [Editor Report · Acceptance letter]

PONE-D-25-18575R1

PLOS ONE

Dear Dr. Ergün,

I'm pleased to inform you that your manuscript has been deemed suitable for publication in PLOS ONE. Congratulations! Your manuscript is now being handed over to our production team.

Kind regards,

on behalf of

Prof. Dr. Onder Aydemir

Academic Editor

PLOS ONE